# Weakly-Supervised Midline Shift Quantification through Simulating the Reversed Disease Progression

**Chih-Chieh Chen**[1]                                                JACKFRANK@GMAIL.COM

**Chang-Fu Kuo**[1,2,3]                                              ZANDIS@GMAIL.COM

[1] *Center for Artificial Intelligence in Medicine, Chang Gung Memorial Hospital, Taoyuan, Taiwan*

[2] *Medical Education Department, Chang Gung Memorial Hospital, Taoyuan, Taiwan*

[3] *Division of Rheumatology, Allergy and Immunology, Chang Gung Memorial Hospital, Taoyuan, Taiwan*

**Editors:** Under Review for MIDL 2025

## Abstract

Medical segmentation masks are often scarce. To get visual and quantitative information, we propose constructing trajectories from anomaly data to normal data using conditional flow matching on an autoencoder, augmented with an auxiliary classification head in the latent space. We demonstrate the effectiveness of our method through weakly supervised midline shift estimation.

**Keywords:** Medical Image Segmentation, Conditional Flow Matching, Weakly Supervised Midline Shift Measurement.

## 1. Introduction

We plan to focus on weakly supervised midline shift quantification using only classification annotations. To extract more meaningful information from a black-box classifier, we propose constructing trajectories through conditional flow matching (Lipman et al., 2023; Liu et al., 2023; Albergo and Vanden-Eijnden, 2023; Tong et al., 2023). In this approach, we select abnormal images as the source distribution and normal images as the target distribution, thereby establishing a coupling from the source to the target distribution. We worked on the latent space of an autoencoder and demonstrate that, our method can not only be served as a visualization tool, but also provide meaningful quantitative information.

## 2. Method

Our method is illustrated in Fig. 1. The flow matching method relies on interpolating samples from both the source and target domains as training data. If we directly use images with midline shifts as the source distribution and images without midline shifts as the target distribution, the model may learn to interpolate between the two images, which is not our goal. We want to reverse the disease process with minimal metamorphosis (Niethammer et al., 2011). Therefore, we propose training a flow matching network in the latent spaces of an autoencoder that is pre-trained on images with and without midline shifts. To further quantify our results, we apply diffeomorphic registration to measure the displacement between the original and transformed images, which serves as an estimate of the midline shift distance of the original image.

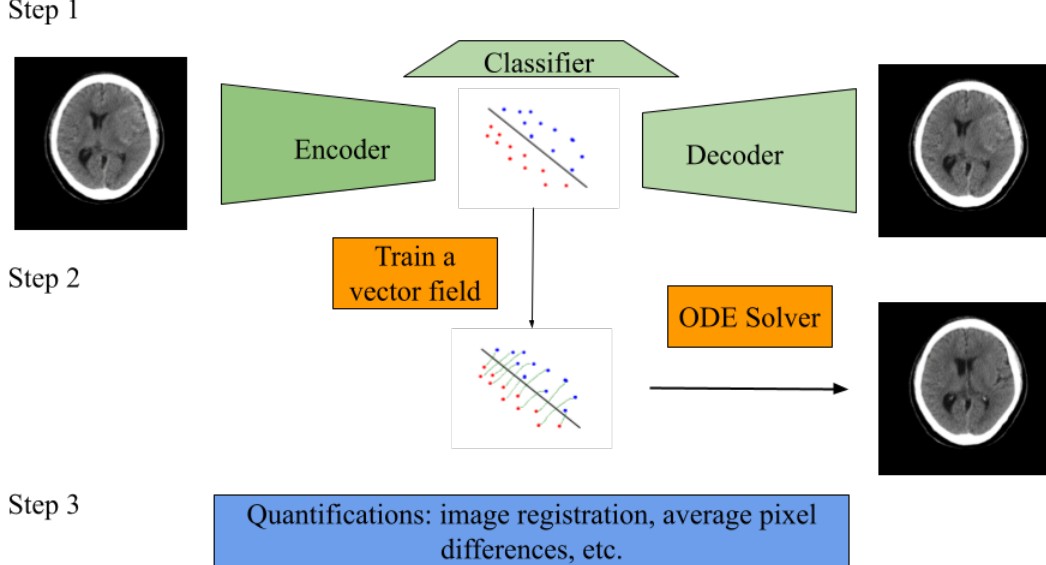

**Figure 1:** Architecture of our proposed network.

## 3. Experiments

For the image autoencoder, we utilized VQGAN (Esser et al., 2021) with an auxiliary linear classification head operating in the latent space. All images were resized to $256 \times 256$ with the latent space having dimensions of 16 (height) $\times$ 16 (width) $\times$ 256 (channel size). We implemented OT-CFM (Tong et al., 2023) and employed a UNet architecture with attention mechanisms in the bottleneck layer to model the trajectory of reversed disease progression. The batch size was set to 70. For image registration, we utilized airlab (Sandkühler et al., 2018). All the experiements were conducted with single Nvidia V100 GPU. The training and testing datasets were from the Chang Gung Research Database (CGRD) (Tsai et al., 2017). For the training data, we used 1177 slices with midline shifts and 2221 normal slices from 907 Brain CT images. During the training of the flow matching network, the slices are further rigid registered to ensure minimal rotation are involved in the reversed disease progression path. For testing data, we used 294 normal slices, 287 slices with midline shift distances ranging from $2mm$ and $5mm$, and 160 slices with midline shift distances greater than $5mm$.

## 4. Results

The visualization of our generated reversed disease progression is shown in Fig. 2. Although we did not jointly train our autoencoder and flow matching model, we were surprised to find that the generated paths appear very realistic. However, when we trained flow matching model with non-registered images, we encountered some unsuccessful generated images, as displayed in Fig. 3. The mean absolute errors (MAE) between the ground truths and measured displacements are presented in Table 1. While the results may not be as

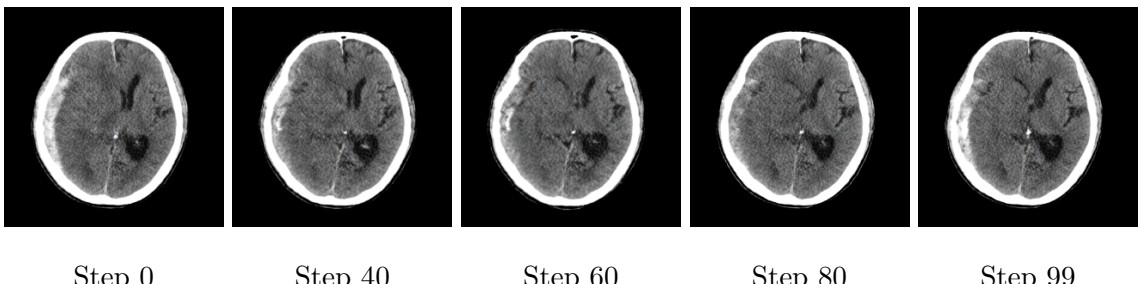

| Step 0 | Step 40 | Step 60 | Step 80 | Step 99 |

**Figure 2:** Reversed disease progression generated by our proposed architecture.

precise as those obtained through fully supervised segmentation methods (Qin et al., 2021), we empirically observed that our model tends to underestimate cases with large midline shifts. We believe the phenomenon may be attributed to the training data size not being large enough for effective generalization of the flow matching/diffusion models (Kadkhodaie et al.) and can be improved by incorporating more data with larger midline shift distances.

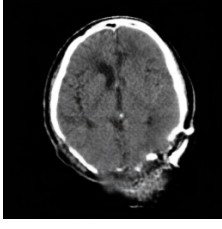

**Figure 3:** Samples if training on non-registered cases.

| MLS (mm) | # of slices | MAE (mm) | Mean of disp. (mm) | STD of disp. (mm) |
|---|---|---|---|---|
| $[0, 2)$ | 294 | 1.19 | 1.54 | 1.98 |
| $[2, 5)$ | 287 | 2.59 | 3.96 | 3.08 |
| $> 5$ | 160 | 4.23 | 5.94 | 2.64 |
| All | 741 | 2.38 | 3.42 | 1.98 |

**Table 1:** Mean absolute errors (MAE) between ground truth midline shift distances and measured displacements.

## 5. Conclusion

When training a neural network with only classification labels, it can be challenging to incorporate more meaningful information. In this study, we propose a novel approach to address this issue by reinterpreting the network's predictions. Specifically, we construct reversed disease paths within the latent space of the classifier. Our empirical results demonstrate that these generated paths are realistic. Additionally, we measure the displacements between the original and generated images to estimate midline shift distances. Our findings indicate a strong correlation between these displacements and the actual midline shift distances. In the future, we plan to explore methods to improve our results, such as increasing the batch size, expanding the training dataset, and making other careful adjustments.

## Acknowledgments

Both authors acknowledge funding from the Center for Artificial Intelligence in Medicine at Chang Gung Memorial Hospital, via grant agreement no. CLRPG3H0016.

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
