# OpenReview forum: "Weakly-Supervised Midline Shift Quantification through Simulating the Reversed Disease Progression"
_MIDL.io/2025/Short_Papers — MIDL 2025 - Short Papers_

### Official Review · Reviewer_icD4 · 2025-04-16

**Rating:** 3
**Confidence:** 4

**Summary:**

This paper introduces a weakly supervised method for midline shift quantification in brain CT images by simulating reversed disease progression. The core idea is to construct trajectories from pathological to normal images in the latent space of a pre-trained autoencoder using conditional flow matching. This allows both visualization and estimation of MLS using only classification labels, avoiding the need for segmentation masks. Displacement between the original and transformed images is used to approximate MLS distance.

**Strengths:**

- The concept of simulating reversed disease progression via CFM in latent space is novel and creative, particularly for weakly supervised settings.
- Operates with only classification labels, addressing a key bottleneck in medical imaging—lack of detailed annotations.
- The generated disease reversal trajectories are compelling and could serve as explainability tools.
- The method shows promising MAE values for MLS estimation without using segmentation ground truth.

**Weaknesses:**

- While promising, the MAEs are relatively high compared to supervised segmentation approaches, especially in the high-shift (>5mm) category.
- Results are based only on internal data from the Chang Gung Research Database; robustness to data distribution shifts remains untested.
- The reasoning behind model choices (e.g., VQGAN, rigid registration) could be more thoroughly justified.
- The method systematically underestimates large MLS values, which could limit clinical applicability unless corrected.

---

### Decision · Program_Chairs · 2025-05-01

Accept